# PAM-Independent Cas12a Detection of Specific LAMP Products by Targeting Amplicon Loops

**DOI:** 10.3390/ijms26168014

**Published:** 2025-08-19

**Authors:** Konstantin G. Ptitsyn, Leonid K. Kurbatov, Svetlana A. Khmeleva, Daria D. Morozova, Olga S. Timoshenko, Elena V. Suprun, Sergey P. Radko, Andrey V. Lisitsa

**Affiliations:** 1V.N. Orekhovich Institute of Biomedical Chemistry, 10 Pogodinskaya St., 119121 Moscow, Russia; konstantin157@yandex.ru (K.G.P.); leonid15@mail.ru (L.K.K.); diny1204@yandex.ru (S.A.K.); darya.d.morozova@gmail.com (D.D.M.); ryzhakova.olga@list.ru (O.S.T.); lisitsa052@gmail.com (A.V.L.); 2Chemistry Faculty, M.V. Lomonosov Moscow State University, 1/3 Lenin Hills, 119991 Moscow, Russia; lenasuprun@mail.ru

**Keywords:** loop-mediated isothermal amplification, Cas12a, amplicon loops, *Clavibacter* species

## Abstract

A straightforward approach is suggested to selectively recognize specific products of loop-mediated isothermal amplification (LAMP) with the Cas12a nuclease without a need for a protospacer adjacent motif (PAM) in the sequence of LAMP amplicons (LAMPlicons). This strategy is based on the presence of single-stranded DNA loops in LAMPlicons and the ability of Cas12a to be *trans*-activated via the binding of guide RNA (gRNA) to single-stranded DNA in the absence of PAM. The approach feasibility is demonstrated on *Clavibacter* species—multiple bacterial plant pathogens that cause harmful diseases in agriculturally important plants. For *Clavibacter* species, the detection sensitivity of the developed PAM-independent LAMP/Cas12a system was determined by that of LAMP. The overall detection selectivity was enhanced by the Cas12a analysis of LAMPlicons. It was shown that the LAMP/Cas12a detection system can be fine-tuned by carefully designing gRNA to selectively distinguish *C. sepedonicus* from other *Clavibacter* species based on single nucleotide substitutions in the targeted LAMPlicon loop. The suggested loop-based Cas12a analysis of LAMPlicons was compatible with the format of a single test tube assay with the option of naked-eye detection. The findings broaden the palette of approaches to designing PAM-independent LAMP/Cas12a detection systems with potential for on-site testing.

## 1. Introduction

Today, the selective amplification of unique nucleic acid sequences in genomes/transcriptomes of pathogenic microorganisms or viruses has become an indispensable tool in DNA diagnostics of infectious diseases and beyond [1]. Although polymerase chain reaction (PCR) still dominates DNA diagnostics and remains the “gold standard” for the field, there is a growing trend towards on-site (out-of-laboratory) testing with numerous applications in medicine, ecology, and agriculture [2,3]. For on-site testing, the isothermal amplification of nucleic acids with simplified instrumentation requirements but reaction times and sensitivity comparable to PCR appears as an attractive approach [4]. Among isothermal amplification methods, loop-mediated isothermal amplification (LAMP) is mostly used, with numerous examples of practical applications [5]. LAMP, introduced in 2000 by Notomi et al. [6], employs a single DNA polymerase with strand-displacing activity and a set of primers (from four to six). This method provides a remarkably high yield of peculiar cauliflower-like DNA amplicons, often referred to as LAMPlicons. LAMPlicons are composed of varying numbers of stem/loop-inverted DNA repeats [5].

Recently, further development in pathogen-specific detection assays for on-site diagnostics has been made by coupling LAMP with the RNA-guided CRISPR/Cas12 analysis of LAMPlicons (CRISPR, clustered regularly interspaced short palindromic repeats; Cas, CRISPR-associated protein) [7]. The interest in CRISPR/Cas12 biosensing systems based on both the functional activity of Cas12 nucleases alone and via coupling them to isothermal amplification is steadily growing [8,9]. By complexing Cas12 nuclease with a particular RNA molecule (guide RNA; gRNA), specific LAMP products can be selectively recognized. The recognition occurs by binding the gRNA segment known as a “spacer” to a complementary sequence (“protospacer”) in double-stranded sections of LAMPlicons and results in the acquisition of *trans*-activity by the Cas12 nuclease. The *trans*-activity is most often detected by an increase in fluorescence due to the nonspecific cleavage of short DNA oligonucleotides labeled with a fluorophore and a fluorescence quencher (molecular reporters, MRs) [7,8,9].

The binding of a gRNA spacer to a target’s protospacer requires the presence of the protospacer adjacent motif (PAM). PAM is a short, specific nucleotide motif recognized by a Cas12 nuclease and positioned at the 5′-end, next to the sequence complementary to the protospacer in the double-stranded DNA (dsDNA) target sequence [10]. The PAM requirement imposes a restriction on the selection of LAMP primers, often making it difficult to integrate the most effective primer set with the CRISPR/Cas analysis of LAMPlicons. To circumvent this limitation, the use of Cas12a mutants and orthologs was suggested to increase the diversity of PAM sequences [11,12]. This approach alleviates the PAM requirement but does not eliminate it. Another way to solve the problem has been recently demonstrated in Ref. [13] by introducing a PAM sequence into a LAMP primer. However, LAMP primers are selected by a LAMP primer-designing software as a “self-consistent” set based on thermodynamic parameters [14], and a change in a sequence of one of LAMP primers can potentially affect the performance of the whole set.

In previous attempts to solve the problem of PAM requirement in LAMP/Cas12 detection systems, the focus was on dsDNA segments of LAMPlicons, which were targeted by a gRNA spacer [11,12,13]. However, LAMPlicons have a loop/stem structure, with loops presented by single-stranded DNA (ssDNA) [5,6]. Advantageously, the Cas12a nuclease, which is mostly combined with LAMP [7,9], can be *trans*-activated by a protospacer sequence in ssDNA in the absence of PAM [10]. Thus, to make the CRISPR/Cas analysis of LAMPlicons truly PAM-independent, it appears straightforward to direct the gRNA spacer to a sequence in a single-stranded loop rather than to that in a double-stranded stem. In the frame of such a strategy, sequences of the so-called “loop primers” could potentially serve as a basis to design gRNA spacers. Indeed, the routinely used software for the selection of LAMP primers can generate loop primers [14] that anneal to ssDNA sequences in the loops and that are employed to speed up the LAMP reaction [15]. To date, the practical feasibility of developing PAM-independent LAMP/Cas detection systems by targeting single-stranded loops of LAMPlicons with a Cas12a/gRNA complex has never been experimentally explored.

The aim of this present work was to test whether a truly PAM-independent LAMP/Cas12a detection system can be developed by targeting single-stranded loops of LAMPlicons with gRNA. The species of *Clavibacter* genus [16] and a set of primers previously designed by Dobhal et al. [17] have been selected as a convenient model for the study. The chosen LAMP primer set allows one to detect all known *Clavibacter* species [17]. The second advantage of the primer set is that it produces LAMPlicons with a loop sequence differing by single-nucleotide substitutions among some *Clavibacter* species. This additionally allowed us to evaluate whether and how different Clavibacter species can be differentiated by fine-tuning the length of gRNA spacers directed to a LAMPlicon loop with nucleotide substitutions. Aside from being a convenient experimental model for our study, *Clavibacter* species are well-known bacterial plant pathogens causing extremely harmful diseases in agriculturally important plants. The diseases include bacterial wilt and canker of tomatoes (*C. michiganensis*), potato ring rot (*C. sepedonicus*), wilting and stunting of alfalfa (*C. insidiosus*), wilt and blight of corn (*C. nebraskensis*), leaf spots and leaf freckles in wheat (*C. tessellarius*), and leaf yellowing in beans (*C. phaseoli*) [16]. *C. michiganensis*, *C. sepedonicus*, and *C. insidiosus* belong to a list of quarantine pathogens in the European Union and some other countries [16]. The potential results of this study could serve as a basis for the development of assays aimed at on-site diagnostics to control these agricultural plant diseases and prevent the pathogens’ spread.

## 2. Results

### 2.1. Sensitivity and Selectivity of LAMP Detection

The overall analytical performance of the LAMP/Cas12a system is determined by that of its constituents, viz., LAMP and the subsequent Cas12a-based detection of specific LAMPlicons. As a first step, the analytical performance of LAMP with the primer set developed by Dobhal et al. for the detection of all *Clavibacter* species [17] was assessed under our experimental conditions in terms of limit of detection (LOD). For that, serial 10-fold dilutions of *C. sepedonicus* genomic DNA (strain As-1405; Appendix A) were used, allowing for loads of 37 pg down to 3.7 fg of genomic DNA per a 10 µL LAMP reaction. Since the size of the *C. sepedonicus* genome is 3.4 × 10^6^ base pairs (bp) [18], that would correspond to the loads of 10^4^ copies to one copy of a bacterial genome per reaction. The number of genomes was calculated based on the mass of a single genome m equal to about 3.7 fg. The mass of a single genome was estimated as follows: m (g) = 650 × 3.4 × 10^6^/*N*_A_, where 650 is the average molecular weight of bp (g/mol) and *N*_A_ = 6.02 × 10^23^ mol^−1^ is Avogadro’s number. The representative amplification curves are demonstrated in Figure 1a. The *C. sepedonicus* genomic DNA was consistently detected down to the load of 37 fg (10 genome copies) per reaction within the time interval of 30 min when the full set of LAMP primers (including loop primers) was used (Appendix A). For other *Clavibacter* species under study (*C. michiganensis*, *C. insidiosus*, *C. nebraskensis*, *C. tessellarius*, and *C. phaseoli*; Appendix A), the load of 37 fg of genomic DNA per LAMP reaction also consistently resulted in an occurrence of sigmoidal amplification curves with approximately the same time of fluorescence rise (exemplified in Figure 1b). Similar results for *C. sepedonicus* were obtained when 200 ng of potato DNA were additionally present in the LAMP reaction (Appendix A). For purposes of comparison, amplification kinetics were quantified by using values of a quantification cycle (*C*_q_), automatically provided by the PCR machine used. Since LAMP amplification curves are usually shown as a change of fluorescence in time, *C*_q_ was converted into “characteristic amplification time” (*t*_c_) based on the cycle duration (0.5 min). No statistically significant differences (*p* ≥ 0.103) were found between mean *t*_c_ values obtained for the same loads of bacterial DNA in the absence and presence of potato DNA (Appendix A). The mean *t*_c_ values gradually decreased with the DNA load both in the absence and presence of potato DNA. However, statistically significant differences were found only between the maximal (10^4^ copies) and two minimal (10 and 100 copies) DNA loads in the presence of potato DNA (*p* = 0.042 and 0.049, respectively). In the absence of potato DNA, all mean *t*_c_ values were found to be statistically indistinguishable (*p* ≥ 0.106).

By extending the reaction time up to 60 min, a typical sigmoidal amplification curve was observed for a load of 3.7 fg (one copy of *C. sepedonicus* genome) per reaction in two out of five repetitions, both in the presence and absence of 200 ng of potato DNA. A similar result was also obtained at that DNA load for other *Clavibacter* species under study. When the reaction time was increased up to 90 min for the load of 3.7 fg of *C. sepedonicus* DNA per reaction, the result was the same: in three repetitions out of five, no sigmoidal amplification curves were observed. The observed irreproducibility in the occurrence of a sigmoidal fluorescence increase even within the extended 90 min reaction time interval makes the detection of bacterial DNA at the load of 3.7 fg unreliable. Consequently, 10 genome copies have to be taken as LOD for the detection of *Clavibacter* species with LAMP under our experimental conditions (for reaction times of up to 90 min). No sigmoid-like increase in fluorescence was observed for any template control of LAMP (NTC LAMP, an aliquot of 1× LAMP buffer instead of a DNA sample) in all cases.

To evaluate LAMP selectivity under our experimental conditions, species of other genera were tested. No sigmoid-like increase in fluorescence within the amplification time of 30 min was observed for *Dickeya*, *Escherichia*, or most *Pectobacterium* species (Appendix A), even at a relatively high load of 37 pg per reaction. However, for *P. odoriferum*, the beginning of a sharp increase in fluorescence was observed. When the amplification time was extended to 60 min, the sigmoidal amplification curves were observed for a number of *Pectobacterium* and *Dickeya* species at that load (Appendix A).

As is known, a LAMP reaction is able to proceed without loop primers but can be sped up by their presence [15]. Indeed, in the absence of loop primers LF and LB (Appendix A), the amplification was about two to three times slower (in terms of *C*_q_ values; exemplified in Appendix A). The sequence of the LAMPlicon loop targeted by primer LB (loop B) was identical in the studied *Clavibacter* species. At the same time, the sequence of loop F (the loop targeted by primer LF) for *C. sepedonicus* differed from that in other *Clavibacter* species by two to three single-nucleotide substitutions (Table 1). For *C. tessellarius*, the sequence of the “loop F annealing site” (the section of loop F to which LF anneals) differs by a single nucleotide from those in other *Clavibacter* species, except for *C. sepedonicus* (Table 1). However, these differences did not result in an appreciable discrepancy of amplification times (Figure 1b) and thus did not allow us to reliably differentiate *Clavibacter* species among themselves with LAMP alone.

### 2.2. The Loop-Targeted Cas12a Analysis of LAMP Products

Figure 2a shows the representative kinetic curves for a cleavage of FAM-MRs (FAM-labeled molecular reporters; Appendix A) by the Cas12a nuclease from the bacterium *Francisella novicida* (FnCas12a). The nuclease was *trans*-activated by LAMP products (1 μL of completed LAMP reaction). The products were recognized by gRNA-B, which targets loop B and is designed based on the sequence of primer LB (Table 1, Table 2 and Appendix A). As can be seen, the increase of fluorescence was observed for LAMP products of all species of the *Clavibacter* genus under study. Also, LAMPlicons generated by both using and not using loop primers were recognized by the FnCas12a/gRNA-B complex (Figure 2b). Although the kinetic curves are provided for LAMP products generated at a load of 37 fg of *Clavibacter* genomic DNA per reaction (Figure 2a), the results can be extended to higher loads as well. Indeed, amplification curves (exemplified by Figure 1) demonstrated approximately equal levels of fluorescence for the DNA-intercalating fluorescent dye EvaGreen at the reaction saturation plateau. The close levels of fluorescence suggest that nearly the same number of LAMPlicons are generated by the end of the LAMP reaction, regardless of DNA load.

In contrast to the *Clavibacter* species, LAMP products generated with genomic DNA from the tested species of other genera (at the extended amplification time, as exemplified in Appendix A) were not recognized by the FnCas12a/gRNA-B complex as specific LAMPlicons (Figure 2a). Similar results were obtained for non-specific LAMP products generated in the absence of loop primers. In all cases, no appreciable rise of fluorescence was observed for the controls (aliquots of LAMP buffer) and NTC LAMP. It is worth mentioning that non-specific LAMPlicons produced from the genomic DNA of *Dickeya* or *Pectobacterium* species and discriminated by the Cas12a analysis (Figure 2a) cannot be unambiguously distinguished from specific LAMPlicons of *C. sepedonicus* by the melting curve analysis (Appendix A).

To demonstrate the utility of the suggested approach for the detection of bacteria in potato tuber tissue, DNA was extracted from potato samples artificially contaminated with *C. sepedonicus*. The extracted DNA was examined both with the commercial real-time PCR test for *C. sepedonicus* detection and with the developed PAM-independent LAMP/Cas12a system. There was a complete concordance between the PCR results and the results obtained with the PAM-independent LAMP/Cas12a detection system (Appendix A). The load of 2.5 genome copies per LAMP reaction was below the LOD value for the developed LAMP/Cas12a detection system (equal to 10 *C. sepedonicus* genome copies per reaction).

The sequence of loop F in the LAMPlicons generated with *C. sepedonicus* genomic DNA is expected to differ from that of other *Clavibacter* species under study (based on sequences retrieved from NCBI databases; Table 1). To examine whether this difference can be utilized to differentiate *C. sepedonicus* from other *Clavibacter* species by means of the loop-based Cas12a analysis of LAMPlicons, a series of gRNAs was designed. The gRNA spacers varied in length and were complementary to sequences in loop F of *C. sepedonicus* LAMPlicons (gRNA-F; Table 1 and Table 2). Figure 3 shows the results of the LAMP product analysis using the FnCas12a nuclease in a complex with gRNA-F-20 (gRNA with a 20 nucleotide (nt)-long spacer designed in part from the sequence of primer LF; Table 1, Table 2 and Appendix A). The complex can specifically recognize LAMPlicons produced from *C. sepedonicus* genomic DNA that is manifested by notably different kinetics of MR cleavage for *C. sepedonicus* and other *Clavibacter* species. No appreciable differences in the rise of fluorescence were observed for NTC LAMP, control, and LAMP products generated with genomic DNA from species of other genera, similar to the FnCas12a/gRNA-B complex (Figure 2a).

For convenience of presentation, the differences in MR cleavage kinetics for *Clavibacter* strains are further analyzed using the initial cleavage rate V_0_. The initial cleavage rate is defined as a slope of an approximately linear segment in the beginning of a kinetic curve (the slope was obtained by using a linear regression fit). When the spacer length was extended to 22 and 24 nt, the selectivity of Cas12a analysis for *C. sepedonicus* detection seemed to worsen (Figure 4). In contrast, by decreasing the length of the spacer to 16 nt, the selectivity can be improved. *C. sepedonicus* can be more evidently distinguished from other *Clavibacter* species by Cas12a analysis with gRNA-F-16 (16 nt-long spacer; Table 2) as seen in Figure 4a and Appendix A. The FnCas12a nuclease in a complex with gRNA-F-14 (14 nt-long spacer; Table 2) was unable to recognize the protospacer and be *trans*-activated. It is worth noting that a shortening of the spacer length from 20 to 16 nt steadily decreased the degree of *trans*-activation of FnCas12a nuclease as manifested by the slower MR cleavage. Still, even at that low cleavage rate, the reliable instrumental detection of the differences was possible (Figure 3, Figure 4, and Appendix A; Appendix A). Also, from a statistical point of view, the mean V_0_ values for *C. sepedonicus* strains Ac-1405 and Ac-2753 differ from those for the tested strains of other *Clavibacter* species for all gRNA-F variants used (*p*-values were less than 0.05; Appendix A). A single exception was observed for the Cas12a/gRNA-F-22 complex, which was unable to statistically differentiate strain Ac-2753 of *C. sepedonicus* from *C. nebraskensis* by the V_0_ value (*p* = 0.164; Appendix A). Another interesting (though expected) observation is that three mismatches in the spacer/protospacer duplex in the case of *C. tessellarius* (Table 1) had a higher impact on the cleavage kinetics than two mismatches if gRNA-F-20 was used. This is apparent by comparing cleavage kinetics and V_0_ values (Figure 3 and Figure 4c). The mean V_0_ value for *C. tessellarius* was the lowest among the *Clavibacter* species tested. The same was kept for the 18 nt-long spacer but not for the 16 nt-long one (Figure 3 and Figure 4a), thus indicating a complex interplay between the spacer length and the number and positions of mismatches.

### 2.3. Visual Detection of C. sepedonicus in a Single Reaction Tube

To demonstrate that the developed PAM-independent LAMP/Cas12a detection system is compatible with a single reaction tube format, the following procedure was adopted. First, LAMP was conducted for 45 min in a 0.2 mL PCR tube using a 10 µL reaction volume (under a layer of mineral oil in a dry block thermostat, which simplifies instrumentation requirements). Then, upon LAMP completion, the tube was carefully opened (so as not to disturb the LAMP reaction volume preserved under the oil layer) and 50 µL of the FnCas12a reaction mixture was placed inside the tube lid. The tube was closed and then hand-shaken or shortly centrifuged to combine the volumes.

However, in such a setup, no *trans*-activation of the FnCas12a nuclease was initially observed. As a first step, the volume of the completed LAMP reaction combined with the Cas12a reaction mixture was systematically varied from 1 µL to 10 µL. It was found that FnCas12a *trans*-cleavage activity does gradually decrease to zero with the added volume of the LAMP reaction. However, if 1 µL of the completed LAMP reaction was combined with 10 µL of the full LAMP reaction mixture (including primers, targets, dNTPs, buffer, etc., but not subjected to incubation at 65 °C), the FnCas12a nuclease retains the ability to acquire the *trans*-cleavage activity. The cleavage kinetics were practically undistinguishable from that for 1 µL of the completed LAMP reaction alone. Thus, the components of the LAMP reaction mixture, per se, do not seem to affect the *trans*-cleavage activity of FnCas12a. Also, the *trans*-cleavage kinetics were practically identical regardless of whether the mineral oil was present or not. Apparently, a reason for the observed FnCas12a inhibition at elevated volumes of completed LAMP reactions may be some bypass reaction products.

A well-known bypass product of enzymatic DNA synthesis is pyrophosphate, which is generated in relatively large amounts by the end of a LAMP reaction [19]. Pyrophosphate binds magnesium ions, which are known to be essential for Cas12a *trans*-activity [20]. However, pyrophosphate is already in the complex with Mg^2+^ by the end of the LAMP reaction [19], and there should be no free pyrophosphate to bind magnesium ions present in the Cas12a reaction mixture. Nevertheless, when we increased the Mg^2+^ concentration in the Cas12a reaction mixture from (commonly used) 6 mM up to 18 mM, the *trans*-cleavage activity was restored (Appendix A). In our experimental setup (50 µL of Cas12a reaction mixture and 10 µL of completed LAMP reaction), the FnCas12a *trans*-cleavage activity apparently reaches a plateau at magnesium concentrations of 12 mM and above (Appendix A). Presently, we do not have enough experimental data to suggest a reasonable hypothesis that would account for the observed inhibition of Cas12a *trans*-cleavage activity and its restoration by elevated magnesium chloride concentrations. Regardless of mechanistic reasons, the found solution was technically simple and allowed us to evaluate the compatibility of the loop-targeted LAMP/Cas12a system with a format of a single test tube detection. The magnesium concentration of 18 mM was chosen as the working one for further experiments to ensure that magnesium ions are present in excess.

Clearly, on-site testing could benefit from a visual (naked eye) detection of MR cleavage by *trans*-activated Cas12a nuclease. The FAM-labeled MRs are most often employed for both instrumental and naked-eye detection [7,9]. However, in the case of naked-eye detection with FAM-labeled MRs, a source of either ultraviolet or blue light is usually needed to illuminate the test tubes. To further simplify the instrumentation requirements, naked-eye detection was carried out by employing a colorimetric assay with ROX-labeled MRs (Appendix A). The cleavage of such reporters by Cas12a nuclease results in a change of color from blue to purple that can be observed simply under daylight [21] and captured with a smartphone camera. As seen in Figure 5, the concentration of ROX-MR-5 (the 5 nt-long ssDNA reporter identical to that of FAM-MR except for the fluorophore and quencher; Appendix A) in the Cas12a reaction mixture is crucial for effective visual detection. The elevated concentration of 18 µM resulted in a clearly visible change of color (Figure 5).

However, the time required for the color development under such conditions was unsatisfactorily long (more than an hour). To further optimize the naked-eye detection conditions, the concentration of Cas12a/gRNA-B complexes was varied, and ROX-MRs of different lengths were tested (ROX-MR-5 and ROX-MR-8 that are 5 and 8 nt long, respectively; Appendix A). The elevation of the Cas12a/gRNA-B concentration in the reaction mixture by 3-fold led to about a 3-fold increase of V_0_ (although at the expense of a rise in the background fluorescence level; Appendix A). Figure 6 shows the color change of the reaction mixture with the elevated FnCas12a concentration (180 nM) and 18 µM of either ROX-MR-5 or ROX-MR-8 at various incubation times. For ROX-MR-8, the color change is already obvious after 5 min of incubation, while for ROX-MR-5—only after 45 min. Interestingly, by that time, the color of the tube with ROX-MR-8 and NTC LAMP became practically indistinguishable from that of the tube where LAMP was conducted with *C. sepedonicus* genomic DNA (Figure 6). LAMP products generated with the DNA of non-target species (Appendix A) exhibited the timing of color development similar to that for NTC LAMP.

The LAMP/Cas12a analysis in a single test tube format under optimized conditions of naked-eye detection was applied to discriminate *C. sepedonicus* from other *Clavibacter* species under study. The Cas12a/gRNA-F-20 complex and ROX-MR-8 were used to perform the analysis. As seen in Figure 7, the samples of LAMP conducted with *C. sepedonicus* genomic DNA do develop a purple color 5 min after being combined with the Cas12a reaction mixture. At the same time, no obvious transition from blue to purple was observed after 5 min for samples of LAMP conducted with DNA extracted from *Clavibacter* species other than *C. sepedonicus* or for NTC LAMP and the control (Figure 7). It should be noted that the fluorescence intensity approached a plateau after 5 min of incubation for samples with *C. sepedonicus* LAMP products (exemplified in Appendix A). For other *Clavibacter* species tested, as well as for NTC LAMP and the control, values of fluorescence intensity were more than two times lower at the plateau level at that moment. However, they all reached plateau levels close to the plateau level for *C. sepedonicus* after 40 min of incubation (Appendix A). Clearly, the color differences are transient, and there is a limited time window to evaluate the reaction outcomes by color transition. However, the use of smartphones equipped with digital cameras makes this a rather easy task to document the observed color differences at selected moments for further comparison and analysis.

## 3. Discussion

The requirement for PAM in an amplicon sequence can constitute a significant obstacle to the development of assays relying on coupling isothermal amplification with a Cas12a-based analysis of amplicons [7,8,9]. The consensus for the PAM sequence for Cas12a nucleases is “TTTN” [22], and the probability of finding such a sequence at any given position in a genome is about 1%, assuming that the distribution of nucleotides is random and uniform. The PAM requirement imposes additional restrictions on selected LAMP primer sets and may not allow one to employ the most effective primers when developing a LAMP/Cas12a-based assay. As mentioned in the introduction, the problem can be alleviated by using engineered Cas12a mutants [11,12]. However, presently, they are not easily available to the research community interested in developing innovative LAMP/Cas12a-based diagnostic tests. The approach relying on the introduction of a PAM sequence into LAMPlicons by modifying LAMP primers [13] appears to be more universal. However, LAMP is known to be prone to nonspecific amplification, mostly due to the larger number of LAMP primers (four to six). The large number of primers increases the probability of undesirable primer/primer interactions [23]. The modification of primers by inserting additional sequences can further heighten the chance of such primer/primer interactions. In contrast to the strategies realized in Refs. [11,12,13], the loop-targeted Cas12a analysis of specific LAMPlicons seems to also be universal. However, the loop-targeted analysis does not require a potentially harmful modification of the otherwise effective LAMP primer set. Furthermore, the design of gRNA is simplified since sequences of loop primers can be utilized as a starting point for the selection of a gRNA spacer sequence.

One might expect that loop primers, after annealing to complementary sections of LAMPlicon loops and extension, can convert ssDNA into dsDNA. As a result, recognition by the FnCas12a/gRNA complex could become impossible since it would require the presence of PAM [10]. However, in most LAMP reaction mixtures (including the one used in our study), the concentration of loop primers is two times lower than those of inner primers FIP and BIP [23]. The LAMP reaction seems to proceed further after the loop primers are exhausted. As a result, a significant number of LAMPlicons with single-stranded loops are always present among LAMP products, allowing for the PAM-independent activation of the Cas12a nuclease (Figure 2b). Moreover, LAMP can be conducted without loop primers, although at the expense of a longer amplification time [23].

In general, one purpose of integrating LAMP with the Cas12a-based analysis of LAMPlicons is to enhance detection selectivity by additionally targeting a DNA sequence different from sequences of primer-annealing sites [7,8,9]. In our experimental setup, annealing sites for loop primers were chosen as protospacers when designing gRNA. It should be noted that LAMP with loop primers, per se, was unable to reliably differentiate *Clavibacter* species among themselves (Figure 1b) despite the presence of nucleotide substitutions in the annealing site for the loop primer F (Table 1). Apparently, the selectivity of LAMP is governed by primers F3, B3, FIP, and BIP, while the loop primers serve to simply speed up the amplification. The loop-targeted Cas12a-based analysis of LAMPlicons obviously enhanced the detection selectivity, allowing us (1) to discriminate *Clavibacter* species from some species of other genera, which produced non-specific amplification with the used LAMP primer set, and (2) to reliably distinguish *C. sepedonicus* from other *Clavibacter* species based on nucleotide substitutions in one of the LAMPlicon loops. As to the former, it is thought that the annealing of LAMP primers to off-target sequences, although imperfect, can still generate non-specific LAMP products at large loads of non-target genomic DNA. These products differ in loop sequences from specific LAMP products and, consequently, are unrecognizable by the Cas12a/gRNA complex. It appears that the previously reported selectivity of the used LAMP primer set [17] was achieved by properly adjusting the amplification time. However, such an approach can potentially pose a risk of false positives in practical applications if non-target microorganisms are present in a tested sample in large amounts.

Also, our findings demonstrate that the selectivity of the loop-targeted LAMP/Cas12a detection system can be further enhanced by appropriately adjusting the length of the gRNA spacer (Figure 3 and Figure 4). For Cas12a nucleases, the gRNA spacer length is typically 20–24 nt [24]. However—only with spacers of 20 nt to 16 nt in length—the selective detection of *C. sepedonicus* was achieved by targeting the loop with nucleotide substitutions. The best result was observed for gRNA-F-16 (16 nt-long spacer). It is known that mismatches in the spacer/protospacer duplex can alter the functional activities of Cas12a nucleases [25,26]. For FnCas12a, a mismatch in the seed region (the first 8–10 nt at the 3′-end of gRNA) was reported to reduce its on-target (*cis*-) cleavage activity but also to increase the *trans*-cleavage activity [27]. The effect of more than one mismatch in the spacer/protospacer duplex on Cas12a *trans*-cleavage activity depends on a complex and presently unpredictable manner of the number of mismatches and their positions [28,29,30]. For a given number of mismatches, the spacer length and even the spacer sequence can dramatically alter the level of Cas12a *trans*-cleavage activity [28,29,30]. As a result, the fine-tuning of gRNA has to be performed on a case-by-case basis. In our case, the simple shortening of the gRNA spacer allowed us to specifically distinguish *C. sepedonicus* from other *Clavibacter* species.

Clearly, the loop-targeted Cas12a analysis of LAMPlicons has the potential for fine-tuning the detection selectivity by systematically varying the priming sites for the gRNA spacer in the loop. As a result, the sequences of the spacer/protospacer duplex and the number and positions of mismatches will change. Although not experimentally tested in this present study, such an approach appears possible and straightforward since the priming of the gRNA spacer on ssDNA is not tied to a PAM location. In contrast, the use of Cas12a mutants [11,12] or the introduction of a PAM sequence into a LAMP primer [13] ties the priming of the gRNA spacer to a location of either natural or artificial PAMs. However, to practically realize greater flexibility in the choice of the gRNA spacer sequence, significant efforts will be required to carefully examine numerous variants of gRNA. These variants have to cover a considerably large variety of sequences, differing in length and by priming positions of the gRNA spacer on the loop, to find the best ones. It would be interesting to see whether artificial intelligence and machine learning could be effective in selecting such variants in the future. However, to do so, there has to be a large body of experimental data available on how the number and positions of mismatches in various sequences impact the acquiring of *trans*-cleavage activity by CRISPR/Cas12 nucleases. Presently, such data are quite sparse [28,29,30].

Aside from selectivity, another important characteristic of analytical performance is LOD, since LAMP produced approximately the same amount of LAMP products at various loads of genomic DNA for the subsequent Cas12a-based analysis of LAMPlicons; namely, LAMP determines the LOD of the LAMP/Cas12a detection system. Rigorously, LOD is defined as the minimal level of an analyte that can be reliably detected [31]. It is common to calculate LOD as a ratio of 3.3× standard deviations for a blank to a slope of the initial linear section of the calibration curve [31]. However, we observed a weak dependence of the *t*_c_-reciprocal on the target amount over the tested range of DNA loads and their significant scatter at the lower end of the range (Appendix A). The *t*_c_-reciprocal, 1/*t*_c_, appears to be a more convenient function of target concentration than *t*_c_ for the purpose of the discussion since it increases with the concentration. Such behavior was not surprising and is common for LAMP, which is known to be a qualitative method rather than a quantitative one, suitable for applications where a simple qualitative result is sufficient [23]. So, it was hardly possible to reliably determine a linear range of the calibration curve. Moreover, the *t*_c_ value for the blank (no template control) is undefinable. Although—for the blank sample—the *t*_c_-reciprocal can be set to zero (assuming *t*_c_ = ∞), one cannot exclude that spurious amplification could be observed for the blank sample at the finite time, but rather beyond the preset reaction time (due to, e.g., low-probability primer/primer interactions). Thus, the standard deviation for the blank cannot be determined in practical terms. Nonetheless, based on the LOD definition as a reliably detectable minimal amount of an analyte, we took LOD for LAMP under our experimental conditions as 37 pg of genomic DNA (10 copies of *C. sepedonicus* genomes) per reaction. Indeed, this load of genomic DNA consistently provided a sigmoidal amplification curve within a 30 min reaction time (overall, in 36 LAMP reactions using DNA extracted from various *Clavibacter* species), whereas no amplification was observed within this time period for NTC LAMP in all cases. The LOD of 10 genome copies per reaction is comparable with LODs reported for the detection of *Clavibacter* species in other studies using nucleic acid amplification methods such as PCR, LAMP, or NASBA (nucleic acid sequence-based amplification). The LOD values varied from a few to tens of bacterial genomes or from fewer than one to a few viable bacteria per reaction (when ribosomal RNA was used as a target in NASBA) [32,33,34,35,36,37,38].

It deserves discussing that the LOD value was reported as 1 fg of bacterial DNA per reaction in the original study where the employed set of LAMP primers was designed [17]. According to our calculations, 1 fg of *C. sepedonicus* genomic DNA would correspond to about 0.3 copies of bacterial genomes per reaction. It is thought that consistent detection at such low target concentrations is hardly possible. Indeed, the probability of having a targeted gene in a sample volume added to a test tube should follow the Poisson distribution and would be significantly below unity at such low DNA concentrations, unless the target gene is present in the genome in multiple copies. However, that is not the case for the targeted gene, which codes ABC transporter ATP-binding protein/ABC transporter permease [17]. The gene is present in the *C. sepedonicus* genome as a single copy gene (https://www.ncbi.nlm.nih.gov/nuccore/MZMM00000000, accessed on 10 July 2025). Actually, for 0.3 target copies per reaction on average, the probability of not having the target in a particular tube can be calculated as 0.74, based on Poisson distribution. This means that merely one positive response in four repetitions should be expected. One may assume that the commercial mixture OptiGene Master Mix from OptiGene (Horsham, UK), used in Ref. [17] to perform LAMP, is more effective than our home-made LAMP reaction mixture with recombinant Bst polymerase from Biolabmix. Indeed, the OptiGene Master Mix contains a different strand-displacing polymerase, viz., the OptiGene’s original GspSSD LF DNA polymerase, with improved characteristics (https://www.optigene.co.uk/gspssd-lf-dna-polymerase, accessed on 3 May 2025). However, even in such a case, the low LOD value appears to be unreachable due to purely stochastic reasons.

Another important point for discussion is the design of the single test tube format implemented in this work. Rigorously, a single test tube assay assumes that all reactions are carried out in a test tube without its opening, including the assessment of the reaction outcome. The reason is to prevent the contamination of the work area by amplification products. However, FnCas12a nuclease cannot tolerate the temperature used to conduct LAMP and has to be added post-reaction. Also, the LAMP reaction volume is commonly small (10 µL in our case) and can evaporate at 65 °C when a dry block thermostat with no lid heating is used. As a way to solve both these problems, we employed a technical trick previously widely practiced in PCR. Namely, we covered a LAMP reaction mixture with a layer of mineral oil prior to starting the amplification reaction. After the reaction was completed, the tube was opened, the Cas12a reaction mixture was placed inside the tube lid, and the tube was again closed. Such a procedure allowed us to avoid any pipetting of the reaction volume after LAMP was completed, including a transfer of LAMP products to another test tube. The latter is a major cause of aerosol contamination. Moreover, the top layer of mineral oil was immiscible with the aqueous LAMP reaction mixture and served as a “liquid cap”. Thus, both reactions were subsequently conducted inside a single test tube. The presence of this “liquid cap” ensured that no potential aerosol contamination of the work area by LAMPlicons could occur upon the tube opening.

Though *Clavibacter* species were used, first of all, as a convenient experimental model, the obtained results provide a good ground for the development of practical assays for the on-site detection of *Clavibacter* species as a genus or as a single species. For the latter, the potential for fine-tuning the gRNA spacer by varying its priming site in the loop F may be explored. In particular, we demonstrated that *C. sepedonicus* can be successfully detected in potato tuber samples by combining LAMP and the loop-targeted Cas12a analysis of LAMPlicons with LOD, comparable to that of a commercial PCR kit (Appendix A). This was achieved by simply varying the length of the gRNA-F spacer. Furthermore, the compatibility of the developed loop-targeted LAMP/Cas12a system with testing in the format of a single test tube assay and naked-eye detection under daylight was clearly demonstrated.

This present study adds to the rapidly growing body of research related to the development of CRISPR/Cas12 biosensing systems—in particular, to that of LAMP/Cas12 detection platforms [7,8,9]. However, the field is still in its infancy, and, as far as we are aware, there are no LAMP/Cas12-based assays implemented into routine practice so far. A number of technical and operational challenges have to be overcome for their widespread practical application. The most significant challenge is a trade-off between practicality and complexity. Though such platforms demonstrate, in general, enhanced selectivity and/or sensitivity of detection, this is achieved at the expense of a more complex design compared to LAMP alone. This raises uncertainties about their scalability, reproducibility, and standardization. For the field to mature, all these issues have to be successfully addressed. The suggested approach to circumvent the PAM requirement may give additional flexibility to research interested in overcoming challenges related to the development of LAMP/Cas12a-based practical assays for on-site pathogen detection.

## 4. Materials and Methods

### 4.1. Reagents

Bst polymerase and 10× LAMP buffer (300 mM Tris-HCl, pH 8.9, 50 mM (NH_4_)_2_SO_4_, 0.5 mg/mL bovine serum albumin, 2% Tween-20) were purchased from Biolabmix (Novosibirsk, Russia; https://biolabmix.ru/en/, accessed on 4 May 2025). The DNA oligonucleotides used, including the LAMP primers developed in Ref. [17], were synthesized, and the HPLC purified by Lumiprobe (Moscow, Russia, https://ru.lumiprobe.com) are listed in Appendix A. An amount of 100 mM of dNTPs in water was supplied by Evrogen (Moscow, Russia; https://evrogen.com/). EvaGreen fluorescent dye (20× stock solution in water) was purchased from Lumiprobe. Agar, yeast extract, and casein/peptone were purchased from Becton Dickinson (Franklin Lakes, NJ, USA); glucose was purchased from Fluka (Buchs, Switzerland); and PCR-grade mineral oil, RNase A, and lysozyme were purchased from Merck (Rahway, NJ, USA). Other chemicals used were of ACS grade or higher and were also received from Merck. Deionized (18 MΩ) Milli-Q water was used to prepare the solutions. The recombinant *Francisella tularensis* CRISPR-nuclease Cas12a (FnCas12a) was expressed and purified as described in Ref. [39]. gRNAs (Table 1) were synthesized enzymatically with a “TranscriptAid T7 High Yield Transcription Kit” (Thermo Fisher Scientific, Waltham, MA, USA) in accordance with the manufacturer’s instructions using an equimolar mixture of T7P DNA oligonucleotide and one of the template DNA oligonucleotides (Appendix A), which was purified as seen in Ref. [39].

### 4.2. Bacteria Culturing and DNA Isolation

Nineteen different strains representing species of the genera *Clavibacter* (eight strains; six species), *Pectobacterium* (nine strains; eight species), and *Dickeya* (two strains; two species), as well as a strain of *Escherichia coli*, were used in the study. The species and host names, strain numbers, and strain source, as well as the countries of origin, are listed in Appendix A. *Clavibacter* species were cultivated on an agar medium at 28 °C. The agar composition was 20 g/L of agar, 5 g/L of yeast extract, 10 g/L of casein/peptone, 5 g/L of glucose, and 5 g/L of NaCl, with pH 7.0–7.2. The suspensions of *Clavibacter* species were prepared by harvesting bacteria from the agar plates and resuspending them in sterile distilled water. Other bacteria were grown at 28 °C as suspensions in LB medium (Dia-M, Moscow, Russia). Bacteria were pelleted by centrifugation, and DNA was extracted with an “innuPREP Bacteria DNA Kit” (IST Innuscreen GmbH, Berlin, Germany) in accordance with the manufacturer’s manual (including lysozyme and RNase A treatments). To extract potato DNA, 100 mg of potato tuber tissue and a spin column-based “SKYSuper Plant Genomic DNA” isolation kit (SkyGen, Moscow, Russia; https://www.skygen.com/) were used. The DNA isolation was carried out following the manufacturer’s protocol. The potato samples artificially contaminated with *C. sepedonicus* were prepared by spiking 100 mg potato tuber tissue samples with 10 µL of 10-fold dilutions of *C. sepedonicus* suspension in sterile distilled water, followed by DNA extraction with a “SKYSuper Plant Genomic DNA” isolation kit. Since *C. sepedonicus* is a Gram-positive bacterium, the extraction procedure was supplemented with lysozyme treatment, similar to that as recommended for the “innuPREP Bacteria DNA Kit”. DNA concentrations were measured on a Qubit fluorimeter (Thermo Fisher Scientific) with a Qubit dsDNA BR Assay kit (Thermo Fisher Scientific), and DNA preparations were aliquoted and stored at −20 °C until further use. The number of bacterial genomes in DNA preparations from *C. sepedonicus*-contaminated potato samples was determined with real-time PCR on a DTprime5 thermal cycler (DNA-Technology, Moscow, Russia) with a commercial kit for *C. sepedonicus* detection in plant tissue (“*Clavibacter michiganensis* subsp. *sepedonicus*-PB”; Syntol, Moscow, Russia) according to the kit’s instruction and using the standard curve kindly provided by the manufacturer.

### 4.3. LAMP Procedure

LAMP was conducted in 1× LAMP buffer, supplemented with dNTPs (1.4 mM each) and 6 mM MgSO_4_. Concentrations of outer (F3 and B3), inner (FIP and BIP), and loop (LF and LB) primers (Appendix A) developed in Ref. [17] were 0.2 µM, 0.8 µM, and 0.4 µM each, respectively. Amplification was performed for various time periods in a 10 µL reaction volume at a constant temperature of 65 °C using Bst polymerase (0.32 unit/µL). The reaction was carried out either in a DTprime5 thermal cycler equipped with an optical system for the real-time detection of fluorescence or in a dry block thermostat Bio TDB-100 (BioSan, Riga, Latvia). Where necessary, either EvaGreen fluorescent dye (40-fold dilution of the stock solution) in the reaction mixture or a mineral oil layer (5 µL in volume) on the top of the reaction mixture was present.

### 4.4. Real-Time Cas12a Cleavage Assay

To carry out the real-time monitoring of FnCas12a cleavage activity induced by LAMP products, FnCas12a nuclease and one of the gRNAs (Table 2) were mixed at equimolar concentrations of 60 nM, if not indicated differently, in the *trans*-cleavage reaction buffer (40 mM Tris-HCl, pH 8.6, 6 mM MgCl_2_, 1 mM dithiothreitol, 0.4% polyethylene glycol, 0.001% Triton X-100, 40 mM glycine) and incubated for 10 min at room temperature. Afterwards, either fluorescein or rhodamine-labeled MRs (FAM-MR and ROX-MR, respectively; Appendix A) was added and the volume was adjusted to 50 µL with the reaction buffer so that the final concentration of MRs was 1 µM, unless specified otherwise. An amount of 1 µL (or 10 µL where indicated) of the completed LAMP reaction was combined with 50 µL of the FnCas12a reaction mixture, and the fluorescence intensity was measured over time in a DTprime5 thermal cycler at a constant temperature of 37 °C. Where necessary, the concentrations of the FnCas12a/gRNA complex and Mg^2+^ ions in the *trans*-cleavage reaction buffer varied in the range of 60 to 180 nM and 6 to 18 mM, respectively.

### 4.5. A Single Test Tube Assay

To conduct an assay in a single test tube format, LAMP was carried out for 40 min in the dry block thermostat Bio TDB-100 in 10 µL of reaction volume covered with a layer of mineral oil (5 µL) using a 0.2 mL PCR tube. Next, the PCR tube was opened and 50 µL of the FnCas12a reaction mixture was placed inside the tube lid. The tube was closed and hand-shaken or shortly centrifuged to mix the volumes. The MR cleavage reaction was carried out in a dry block thermostat Bio TDB-100. In the latter case, the result of ROX-MR cleavage was monitored by a visual observation of test tubes under daylight. The results were documented with a smartphone camera (Samsung Galaxy S8; Samsung Electronics, Seoul, Republic of Korea).

### 4.6. Statistical Treatment

Microsoft Excel’s statistical functions were used to calculate arithmetic means, standard deviations, and confidence level *p*-values based on the results of three to five independent experiments.

## 5. Conclusions

This present study demonstrates the feasibility of a Cas12a-based analysis of LAMP products in a truly PAM-independent fashion by targeting sections of single-stranded LAMPlicon loops. The specific LAMP products generated both with and without loop primers were shown to be effectively recognized by the FnCas12a/gRNA complex in the absence of PAM. The sensitivity of the LAMP/FnCas12a detection system for *Clavibacter* species was determined in practical terms by that of LAMP, per se, and can be taken as 10 copies of bacterial genomes per reaction. The detection selectivity was enhanced by the loop-targeted Cas12a analysis of LAMPlicons. Furthermore, as shown in the example of *Clavibacter* species, the loop-targeted LAMP/FnCas12a detection system can be fine-tuned by carefully designing the gRNA spacer so as to selectively discriminate one *Clavibacter* species from the others. The discrimination was based on the presence of single-nucleotide substitutions in a targeted LAMPlicon loop. The developed PAM-independent LAMP/FnCas12a detection system can be adapted for use as a single test tube assay with options for both instrumental and visual (under daylight) detection. The overall time of the assay in a single test tube format with visual detection did not exceed 1 h. Our findings relax the selection of LAMP primers compatible with Cas12a analysis of LAMPlicons, allowing one to use the most effective primer sets. The demonstrated approach widens the palette of the present strategies for designing PAM-independent LAMP/Cas12a detection systems with potential for on-site testing.

## Figures and Tables

**Figure 1 ijms-26-08014-f001:**
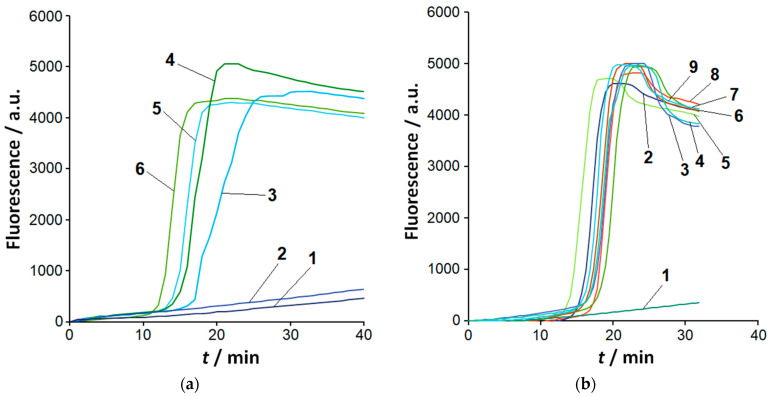
Detection of *Clavibacter* species by LAMP. Representative amplification curves (fluorescence in arbitrary units, a.u.). (**a**) LAMP conducted with *C. sepedonicus* genomic DNA (strain 1405). Curve 1—NTC (no template control); curves 2, 3, 4, 5, and 6—*C. sepedonicus* DNA loads of 3.7 fg, 37 fg, 370 fg, 3.7 pg, and 37 pg per reaction, respectively. (**b**) LAMP conducted with 37 fg of genomic DNA of *Clavibacter* species per reaction. Curve 1—NTC; curves 2 and 3—strains Ac-1403 and Ac-1144 of *C. michiganensis*, respectively; curves 4 to 6—*C. nebraskensis*, *C. phaseoli*, and *C. tessellarius*, respectively; curves 7 and 8—strains Ac-2753 and Ac-1405 of *C. sepedonicus*, respectively; curve 9—*C. insidiosus*.

**Figure 2 ijms-26-08014-f002:**
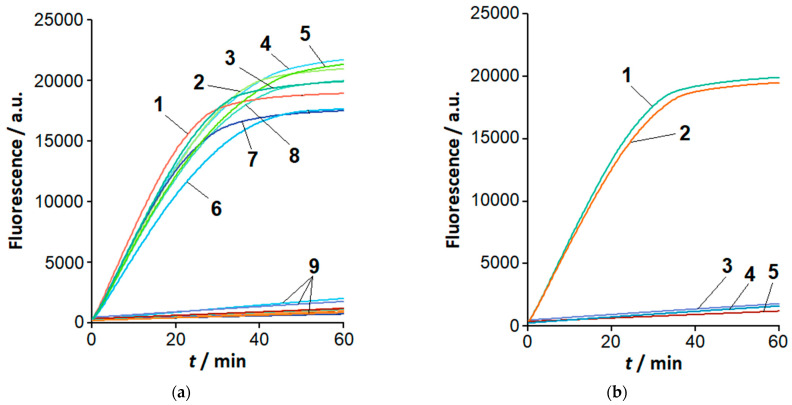
Analysis of LAMP products with the FnCas12a/gRNA-B complex. Representative kinetic curves of FAM-MR cleavage by FnCas12a nuclease. An amount of 1 µL of the completed LAMP reaction, 1 µM of FAM-MR, and 60 nM of FnCas12a/gRNA-B. (**a**) Different *Clavibacter* species. Curves 1 to 8—LAMP products for *C. nebraskensis*, *C. phaseoli*, *C. tessellarius*, *C. insidiosus*, *C. sepedonicus* (strains Ac-2753 and Ac-1405), and *C. michiganensis* (strains Ac-1403 and Ac-1144) genomic DNA, respectively. Curves marked by 9 correspond to species other than *Clavibacter* (LAMP conducted for 60 min with 37 pg of genomic DNA), NTC LAMP, and the control (LAMP buffer instead of completed LAMP reaction). (**b**) The effect of loop primers on the recognition of LAMPlicons by the Cas12a nuclease. Curves 1 and 2—LAMP products generated with and without loop primers, respectively; curves 3 to 5—corresponding NTC LAMP and control. LAMP is the same as seen in Appendix A.

**Figure 3 ijms-26-08014-f003:**
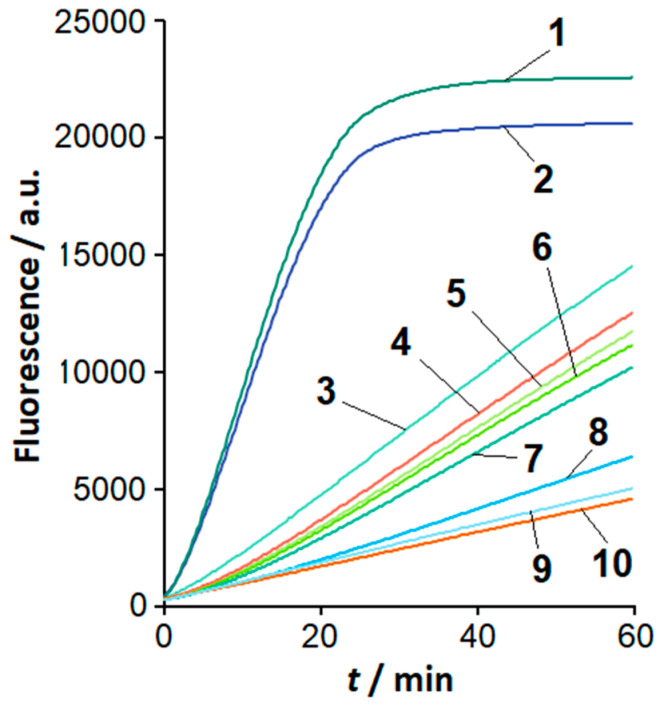
Analysis of LAMP products with the FnCas12a/gRNA-F-20 complex. Representative kinetic curves of FAM-MR cleavage by FnCas12a nuclease. An amount of 1 µL of the completed LAMP reaction, 1 µM of FAM-MR, and 60 nM of FnCas12a/gRNA-F-20. Curves 1 to 10—*C. sepedonicus* (Ac-2753 and Ac-1405), *C. nebraskensis*, *C. michiganensis* (Ac-1403), *C. insidiosus*, *C. phaseoli*, *C. michiganensis* (Ac-1144), *C. tessellarius*, NTC LAMP, and the control, respectively.

**Figure 4 ijms-26-08014-f004:**
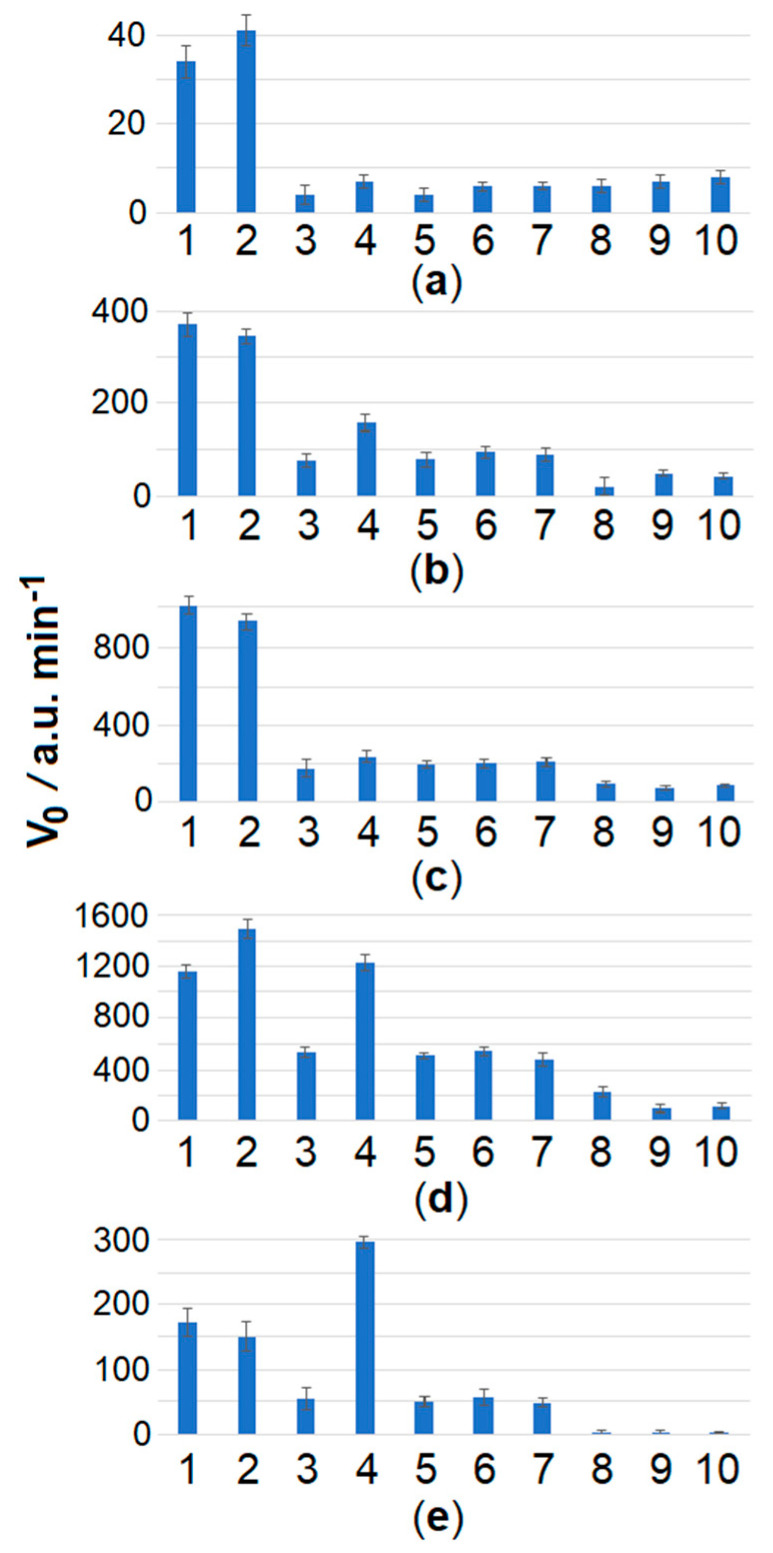
The initial velocity, V_0_, of FAM-MR cleavage using the Cas12a nuclease for different *Clavibacter* species and gRNA-F variants. The LAMP and Cas12a analysis conditions are the same as seen in Figure 3, except for various gRNA-F. Numbers 1 to 10 indicate *C. sepedonicus* (Ac-2753 and Ac-1405), *C. michiganensis* (Ac-1403), *C. nebraskensis*, *C. insidiosus*, *C. phaseoli*, *C. michiganensis* (Ac-1144), *C. tessellarius*, NTC LAMP, and the control, respectively. (**a**) gRNA-F-16, (**b**) gRNA-F-18, (**c**) gRNA-F-20, (**d**) gRNA-F-22, (**e**) gRNA-F-24.

**Figure 5 ijms-26-08014-f005:**
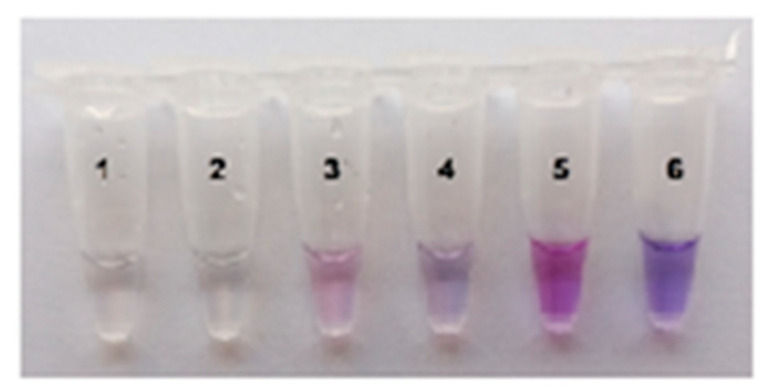
The color development at various ROX-MR-5 concentrations. The single test tube assay with naked-eye detection. Various concentrations of ROX-MR-5: 1 µM—tubes 1 and 2, 10 µM—tubes 3 and 4, 18 µM—tubes 5 and 6. LAMP with 37 fg of *C. sepedonicus* (strain Ac-1405) genomic DNA for 45 min; FnCas12a/gRNA-B complex. Tubes 2, 4, and 6—NTC LAMP.

**Figure 6 ijms-26-08014-f006:**
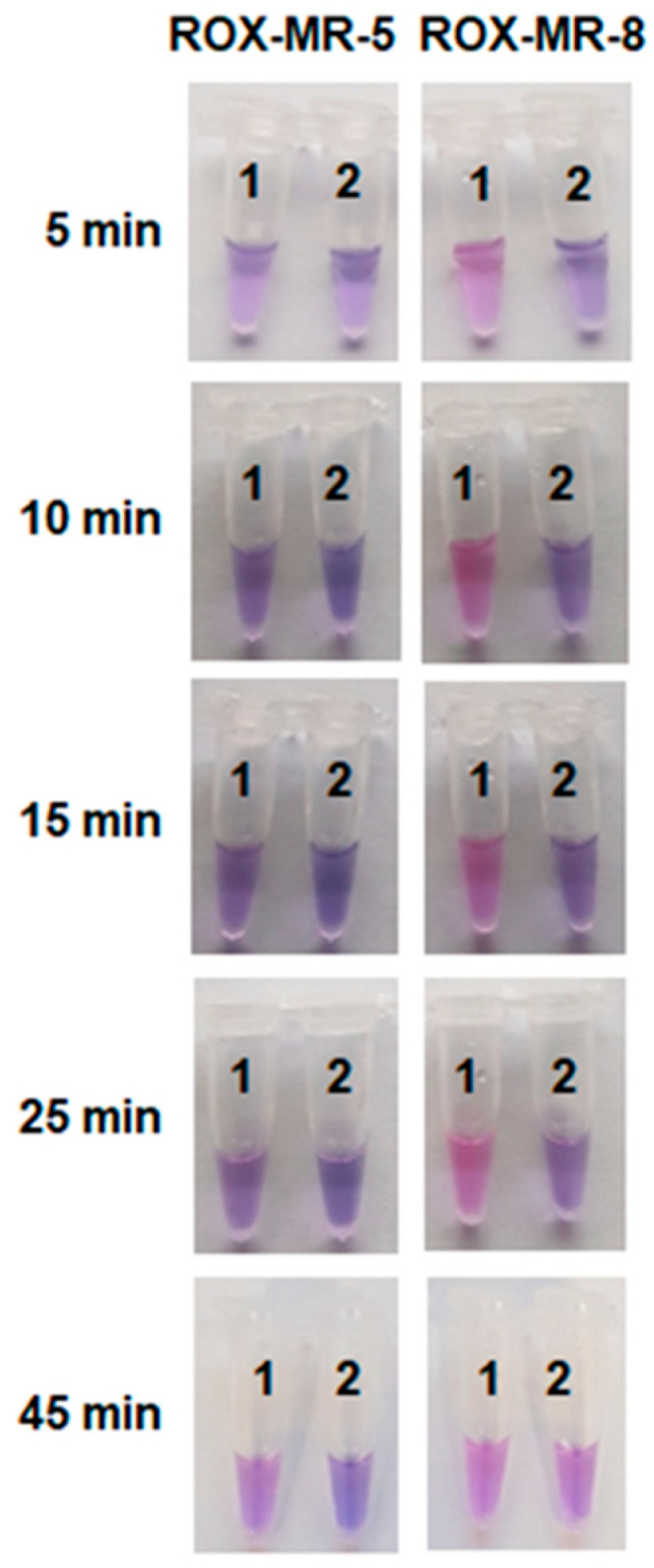
The color development with time. The single test tube assay with ROX-MR-5 or ROX-MR-8 and naked-eye detection. The incubation time is indicated on the left. ROX-MR concentrations—18 µM. Cas12a/gRNA-B complex. Cas12a and Mg^2+^ ions concentration—180 nM and 18 mM, respectively. Tubes 1 and 2—LAMP with 37 fg of *C. sepedonicus* (strain Ac-1405) genomic DNA for 45 min and NTC LAMP, respectively.

**Figure 7 ijms-26-08014-f007:**
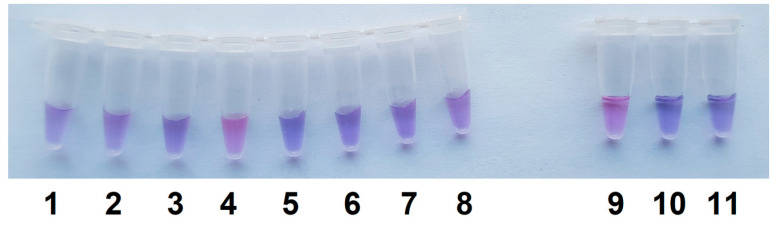
Differential detection of *C. sepedonicus* with the single test tube assay. LAMP is the same as seen in Figure 1b. The other conditions are the same as seen in Figure 6, except that the incubation time was 5 min. Test tubes 1 to 11—*C. insidiosus*, *C. michiganensis* (strains Ac-1403 and Ac-1144), *C. sepedonicus* (strain Ac-1405), *C. tessellarius*, *C. phaseoli*, *C. nebraskensis*, repeat of *C. nebraskensis*, *C. sepedonicus* (strain Ac-2753), NTC LAMP, and the control, respectively.

**Table 1 ijms-26-08014-t001:** Sequences of loops B and F of LAMPlicons. The sequences were retrieved from NCBI databases for *Clavibacter* species based on the sequences of the inner LAMP primers (FIP and BIP; Appendix A). Underlined sections are targeted by gRNA spacers. Annealing sites for primers LF and LB are shown in bold. The single-nucleotide substitutions in the loop F are indicated in red.

Loop Designation	Species	Loop Sequence (5’→3’)
Loop B	All	GGCG**TCTCGCTCCTGAGCCTCCTG**CTCGGGCAGCTCGTCTTCA
Loop F	*C. sepedonicus*	GACTTGCGC**ACGT****TCTCCACGATGATGCG**CG
	*C. tessellarius*	GACTTCCGC**ACGT****GCTCGACGATGATGCG**CG
	*C. michiganensis*	GGACTTCCGC**ACGTTCTCGACGATGATGC****G**CG
	*C. nebraskensis*	GACTTCCGC**ACGTTCTCGAC****GATGATGCG**CG
	*C. insidiosus*	GACTTCCGC**ACGTTCTCGAC****GATGATGCG**CG
	*C. phaseoli*	GGACTTCCGC**ACGTTCTCGACGATGATGC****G**CG

**Table 2 ijms-26-08014-t002:** The list of gRNAs used. The spacer sequence is underlined.

Name	Sequence (5’→3’)
gRNA-B	GGGAAUUUCUACUGUUGUAGAUCAGGAGGCUCAGGAGCGAGA
gRNA-F-14	GGGAAUUUCUACUGUUGUAGAUGGAGAACGUGCGCA
gRNA-F-16	GGGAAUUUCUACUGUUGUAGAUGUGGAGAACGUGCGCA
gRNA-F-18	GGGAAUUUCUACUGUUGUAGAUUCGUGGAGAACGUGCGCA
gRNA-F-20	GGGAAUUUCUACUGUUGUAGAUCAUCGUGGAGAACGUGCGCA
gRNA-F-22	GGGAAUUUCUACUGUUGUAGAUAUCAUCGUGGAGAACGUGCGCA
gRNA-F-24	GGGAAUUUCUACUGUUGUAGAUGCAUCAUCGUGGAGAACGUGCGCA

## Data Availability

Data are contained within the article and Appendix A.

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
