# Peer review of "PAM-Independent Cas12a Detection of Specific LAMP Products by Targeting Amplicon Loops"

_ijms, 2025, doi:10.3390/ijms26168014_

Round 1
Reviewer 1 Report
Comments and Suggestions for Authors
The reviewed manuscript is dedicated to optimization and validation of a novel PAM-free Cas12a-based approach for detection of LAMP products. The topic itself is in a common trend of modern molecular diagnostics towards simplified tests for point-of-care testing and the experimental design is rigorous and carefully planned, supporting the made conclusions. However, several questions listed below need to be addressed before a possible publication.
- The writing style of the manuscript is rigorous but seems to be overcomplicated. Many sentences could be simplified in order to make them clearer for readers and such correction could greatly increase readability of the manuscript. As an example, the following statement in the abstract needs to be clarified: “In regard to Clavibacter species, the detection sensitivity of the developed PAM-independent LAMP/Cas12a system was determined by that of LAMP, while the overall detection selectivity was enhanced by the Cas12a analysis of LAMPlicons.”
- Full species names instead of abbreviations will help better acknowledge the targeted pathogen.
- A figure demonstrating binding sites of primers and probes with a LAMP amplification product will clarify how the suggested Cas12a-based methods actually works.
- Commonly, fluorescent curves of LAMP are analyzed using the same strategy as in qPCR, e.g., a threshold between positive and negative reactions is calculated using first cycles where accumulation of fluorescent signal is invisible. These time-to-threshold values are calculated automatically by a PCR machine software similarly to Cq values in qPCR. Suggested by authors “characteristic amplification time” seems to be susceptible to fluctuation of end-point fluorescence level which can be seen in Figure 1 and other figures. Thus, this metric can be biased and does not always correlate with actual LAMP kinetics. In that light, the LAMP speed could be actually different when template DNA was titrated. Authors are requested to recalculate their data using more conventional approach that was previously validated and approved as a robust assessment of LAMP kinetic.
- The number of technical repeats (5) can compromise the LoD assessment because LAMP efficacy with low template concentrations is highly variable. More reliable estimation can be obtained by analyzing 32 or similar number of replicates for low DNA concentrations close to an assumed LoD.
- Page 7, lines 232–233: “No rise of fluorescence was observed for LAMP products generated with genomic DNA from species of other genera” seems to contradict with Figure 3, where fluorescence is linearly rises for non-targeted species, while for the targeted specie this rise is sigmoidal.
- Speculatively, the Cas12a detection relies heavily on end-point fluorescence instead, making cleavage kinetic a secondary parameter necessary mostly for reaction optimization. If so, presentation of end-point fluorescence is necessary for selection of the best gRNA lengths and the number of mismatches.
- Authors demonstrated non-specific change of color for NTC controls if reactions are incubated for a prolonged time. However, they did not reveal whether was the same effect observed for non-targeted species, or not.
- Line 498: Twin-20 instead of Tween-20.
Author Response
The reviewed manuscript is dedicated to optimization and validation of a novel PAM-free Cas12a-based approach for detection of LAMP products. The topic itself is in a common trend of modern molecular diagnostics towards simplified tests for point-of-care testing and the experimental design is rigorous and carefully planned, supporting the made conclusions. However, several questions listed below need to be addressed before a possible publication.
We thank the Reviewer for the thorough assessment of our work and valuable comments.
- The writing style of the manuscript is rigorous but seems to be overcomplicated. Many sentences could be simplified in order to make them clearer for readers and such correction could greatly increase readability of the manuscript. As an example, the following statement in the abstract needs to be clarified: “In regard to Clavibacter species, the detection sensitivity of the developed PAM-independent LAMP/Cas12a system was determined by that of LAMP, while the overall detection selectivity was enhanced by the Cas12a analysis of LAMPlicons.”
We went through the manuscript and simplified sentences where it seems possible in order to increase readability of the manuscript.
- Full species names instead of abbreviations will help better acknowledge the targeted pathogen.
Full species names instead of abbreviations were used in the revised manuscript as recommended by the Reviewer.
- A figure demonstrating binding sites of primers and probes with a LAMP amplification product will clarify how the suggested Cas12a-based methods actually works.
The requested scheme is provided in the graphic abstract submitted as a part of the manuscript.
- Commonly, fluorescent curves of LAMP are analyzed using the same strategy as in qPCR, e.g., a threshold between positive and negative reactions is calculated using first cycles where accumulation of fluorescent signal is invisible. These time-to-threshold values are calculated automatically by a PCR machine software similarly to Cq values in qPCR. Suggested by authors “characteristic amplification time” seems to be susceptible to fluctuation of end-point fluorescence level which can be seen in Figure 1 and other figures. Thus, this metric can be biased and does not always correlate with actual LAMP kinetics. In that light, the LAMP speed could be actually different when template DNA was titrated. Authors are requested to recalculate their data using more conventional approach that was previously validated and approved as a robust assessment of LAMP kinetic.
The data have been recalculated based on Cq values as requested by the Reviewer. Cq values were automatically provided by the PCR machine used in the study. Since Cq is a dimensionless parameter while amplification curves are presented as “fluorescence vs. time”, for convenience of discussion we converted Cq values into “characteristic amplification times”, based on the cycle duration of 0.5 min. The recalculated data are provided in Table S3. The pertinent changes have been made in the text of the Results section (lines 130-134 and 138-139).
- The number of technical repeats (5) can compromise the LoD assessment because LAMP efficacy with low template concentrations is highly variable. More reliable estimation can be obtained by analyzing 32 or similar number of replicates for low DNA concentrations close to an assumed LoD.
We agree with the Reviewer that the position of S-shaped amplification curve on time axis is very variable for low DNA loads. Our estimate of LOD as 37 pg of genomic DNA (10 copies of C. sepedonicus genomes) per LAMP reaction was based on observation that such load consistently provided a sigmoidal amplification curve within 30 min reaction time, regardless of its precise position on time axis. The justification for such approach is provided in the Discussion section (lines 463-487). Overall, 31 LAMP reactions were conducted in our study with the load of 37 pg, using DNA extracted from various Clavibacter species, and for all cases we observed a sigmoidal amplification curve within 30 min reaction time. Upon revision, we additionally conducted 10 LAMP reactions – 5 with 37 pg of C. sepedonicus and 5 NTCs. Consequently, the overall number of LAMP reactions with the load of 37 pg of genomic DNA has increased up to 36. The number of LAMP reactions with the load of 37 pg of genomic DNA is provided on line 485 of the revised manuscript.
- Page 7, lines 232–233: “No rise of fluorescence was observed for LAMP products generated with genomic DNA from species of other genera” seems to contradict with Figure 3, where fluorescence is linearly rises for non-targeted species, while for the targeted specie this rise is sigmoidal.
The sentence has been corrected to “No appreciable rise of fluorescence was observed for …” (line 245).
- Speculatively, the Cas12a detection relies heavily on end-point fluorescence instead, making cleavage kinetic a secondary parameter necessary mostly for reaction optimization. If so, presentation of end-point fluorescence is necessary for selection of the best gRNA lengths and the number of mismatches.
We thank the Reviewer for raising an important question. However, we kindly disagree with the Reviewer that cleavage kinetics is necessary mostly for reaction optimization. In general, end-point detection assumes that a reaction is complete. However, in practice, end-point detection often means the detection at a particular time point. At that point, the reaction can be intentionally stopped or its temporal outcome can be documented, even without stopping the reaction. In the case of the Cas12a cleavage kinetics, fluorescence is measured either in real time manner (instrumental detection) or at a particular time point (naked-eye detection). For instrumental detection, the measurement of end-point fluorescence is unnecessary, since judgment can be made based on clear differences in cleavage kinetics (Figures 3, 4, and S4). For the naked-eye detection, the visual examination of test tubes at the fifth minute after combining the LAMP reaction volume with the Cas12a/gRNA-B reaction mixture was experimentally found optimal under particular conditions of cleavage reaction, as illustrated by Figure 6. The difference in color at that time point is fully defined by cleavage kinetics and can be considered as “surrogate end-point fluorescence”. As clear from Figure 6, there is a limited time window (from 5 to 25 min or so) for evaluating reaction outcomes since not all cleavage reactions are completed and observed differences are based on differences of reaction rates. However, the outcomes of cleavage reactions can be easily documented at a selected time point with a smartphone camera for further comparison and analysis.
In the present work, we did not attempt to optimize timing of naked-eye “end-point” detection for other gRNAs than gRNA-F-20. As we specifically pointed in the Introduction and Discussion sections, Clavibacter species were employed as a convenient model. The primary aim of the study was to demonstrate the feasibility to circumvent the PAM requirement by targeting single-stranded loops of LAMPlicon rather than their double-stranded stems. Since LAMP/Cas12a detection systems are considered, first of all, as a new approach to development of on-site DNA diagnostics, we demonstrated on the example of gRNA-B and gRNA-F-20 that both instrumental and non-instrumental detection is possible with the loop-targeted Cas12a analysis of LAMP products. Furthermore, we demonstrated that single nucleotide substitutions in single-stranded LAMPlicon loops can be exploited to selectively distinguish one species among other closely related species with PAM-free loop-targeted Cas12a analysis in both instrumental and non-instrumental modes. We did not pursue a goal (and nowhere claimed it in the manuscript) to develop a practical assay for on-site detection of a particular Clavibacter species (e.g., C. sepedonicus). That would indeed require a careful examination of numerous variants of gRNA, differing in length and/or by priming positions of gRNA spacer on the loop, to find the best optimal conditions. We believe that such work is out of scope of the present “proof-of-concept” study.
- Authors demonstrated non-specific change of color for NTC controls if reactions are incubated for a prolonged time. However, they did not reveal whether was the same effect observed for non-targeted species, or not.
The same effect was observed for non-target species as well. In response to the Reviewer’s question, we specified it upon revision (lines 352-353).
- Line 498: Twin-20 instead of Tween-20.
We apologize for such a typo. It has been corrected.
All changes made in the manuscript are highlighted by yellow.

Reviewer 2 Report
Comments and Suggestions for Authors
This research presents an innovative approach, a PAM-independent LAMP/Cas12a detection system, which holds great promise for rapid and specific pathogen detection in the field. Many innovative features, such as the selective detection design using gRNA and the visual output method, are highly appreciated.
However, the following technical and operational challenges remain for the widespread practical application of this method:
-
Limitations of quantitation: LAMP itself is a qualitative method, and quantitative reliability is limited by reaction time and variability.
-
Risk of false positives: Although the gRNA design provides high specificity, the risk of false positives due to mismatches and spurious amplification cannot be ignored.
-
Trade-off between sensitivity and practicality: Complex samples can be susceptible to noise and interference, making a simple, highly sensitive design ineffective.
-
Difficulties in reproducibility and standardization: Optimization of gRNA design and reaction conditions is case-by-case, posing a barrier to standardization and mass production.
-
Uncertainty about scalability: Further challenges remain for practical implementation, including ease of use, cost, stability, and interpretation methods.
Considering these issues, it seems more realistic in terms of widespread adoption and feasibility to first return to a simple, highly sensitive design based on the conventional LAMP method, demonstrate its effectiveness in limited situations, and then gradually improve it.
Furthermore, to expand the scope of application of this method in the future, the introduction of computational support technologies such as automated gRNA target selection and sequence optimization using AI and machine learning would be extremely effective. Comparative analysis using public genome databases and the development of gRNA design support tools incorporating mismatch tolerance and specificity scoring are expected to significantly improve the scalability, reliability, and efficiency of this method.
I strongly recommend that you discuss these technical challenges and future prospects (especially potential development through AI integration) in the Discussion section. I believe that doing so will more clearly convey to readers the applicability and strategic value of this research.
Author Response
This research presents an innovative approach, a PAM-independent LAMP/Cas12a detection system, which holds great promise for rapid and specific pathogen detection in the field. Many innovative features, such as the selective detection design using gRNA and the visual output method, are highly appreciated.
However, the following technical and operational challenges remain for the widespread practical application of this method:
Limitations of quantitation: LAMP itself is a qualitative method, and quantitative reliability is limited by reaction time and variability.
Risk of false positives: Although the gRNA design provides high specificity, the risk of false positives due to mismatches and spurious amplification cannot be ignored.
Trade-off between sensitivity and practicality: Complex samples can be susceptible to noise and interference, making a simple, highly sensitive design ineffective.
Difficulties in reproducibility and standardization: Optimization of gRNA design and reaction conditions is case-by-case, posing a barrier to standardization and mass production.
Uncertainty about scalability: Further challenges remain for practical implementation, including ease of use, cost, stability, and interpretation methods.
Considering these issues, it seems more realistic in terms of widespread adoption and feasibility to first return to a simple, highly sensitive design based on the conventional LAMP method, demonstrate its effectiveness in limited situations, and then gradually improve it.
Furthermore, to expand the scope of application of this method in the future, the introduction of computational support technologies such as automated gRNA target selection and sequence optimization using AI and machine learning would be extremely effective. Comparative analysis using public genome databases and the development of gRNA design support tools incorporating mismatch tolerance and specificity scoring are expected to significantly improve the scalability, reliability, and efficiency of this method.
I strongly recommend that you discuss these technical challenges and future prospects (especially potential development through AI integration) in the Discussion section. I believe that doing so will more clearly convey to readers the applicability and strategic value of this research.
We thank the Reviewer for positive assessment of our work in general and valuable comments. The Reviewer raises very important issues of general interest. We believe that these issues deserve a broad discussion in specialized articles such as a review or an expert opinion. In our manuscript, we presented results of a focused experimental study devoted to a new straightforward approach to circumvent the PAM requirement in LAMP/Cas12a-based detection systems. We do not feel that it would be appropriate to broadly discuss these important issues of general interest in an article of such kind. We have shortly discussed perspectives for designing gRNAs with the use of AI and machine learning (lines 454-462 of the revised manuscript) as recommended by the Reviewer. Also, we have added a new paragraph at the end of the Discussion section (lines 542-554), devoted to general challenges in development of LAMP/Cas12a-based detection systems, to shortly address the issues raised by the Reviewer.
All changes made in the manuscript are highlighted by yellow.

Reviewer 3 Report
Comments and Suggestions for Authors
The manuscript entitled, ‘PAM-independent Cas12a detection of specific LAMP products by targeting amplicon loops’ reported targeted amplicon loops enable PAM-independent Cas12a detection of certain LAMP products. The article should be modified according the following comments:
- The abstract lacks specificity regarding the data presented in the study. It is recommended to highlight key findings or notable data to give readers a clearer understanding of the study's contributions.
- Have negative controls (NTC LAMP) been run in triplicate or more to confirm reproducibility of the lack of fluorescence?
- Could the authors clarify whether any non-specific amplification products were checked via gel electrophoresis or melting curve analysis?
- Could the authors quantify the observed 2–3× delay in amplification in the absence of loop primers—was this based on threshold time (Tt) or endpoint fluorescence intensity?
- Were there any differences in amplification efficiency or product yield when loop primers were excluded?
- Could the authors provide experimental data or references confirming that pyrophosphate accumulation during LAMP is sufficient to sequester Mg²⁺ and inhibit Cas12a activity under their conditions?
- Was the Cas12a activity restored immediately upon Mg²⁺ addition, or was an incubation period required for complex formation?
Author Response
The manuscript entitled, ‘PAM-independent Cas12a detection of specific LAMP products by targeting amplicon loops’ reported targeted amplicon loops enable PAM-independent Cas12a detection of certain LAMP products. The article should be modified according the following comments:
We thank the Reviewer for appraising our work and useful comments.
The abstract lacks specificity regarding the data presented in the study. It is recommended to highlight key findings or notable data to give readers a clearer understanding of the study's contributions.
Unfortunately, the abstract size is limited and must not to exceed 200 words, according to the Author Guide for IJMS. We believed that did our best to present the aim and key funding of the study within this limit. We honestly tried to improve the abstract upon revision in accordance with Reviewer recommendations but failed to do so within the limit of 200 words.
Have negative controls (NTC LAMP) been run in triplicate or more to confirm reproducibility of the lack of fluorescence?
NTC LAMP were run either in duplicates or triplicates in each series of experiments. The overall number of NTC runs exceeded 30, with LAMP reactions conducted for 30 to 90 min. In all cases, no typical sigmoid-like rise of fluorescence was observed for NTCs within these reaction times. However, we could not exclude that spurious amplification can occur with the used set of LAMP primers beyond the longest amplification time tested (90 min).
Could the authors clarify whether any non-specific amplification products were checked via gel electrophoresis or melting curve analysis?
We did not check amplification products with gel electrophoresis. LAMPlicons are a set of DNA fragments with a wide size distribution. Their patterns on electrophoretic gels are complex and it is not easy to unambiguously distinguish specific and non-specific LAMP products unless they result from primer-primer interactions. For the later, the pattern could be quite different. Since we observed no spurious amplification with LAMP primers alone (NTCs) within the tested amplification times (no sigmoid-like rise of fluorescence), there were no LAMP products to analyze via electrophoresis. Besides, electrophoretic analysis increases the probability of contamination of lab area with LAMP products that we tried to minimize. As to the melting curve analysis, it appears to hardly work for LAMP in general, unless some special fluorescent probes (e.g., https://doi.org/10.1038/s41598-017-04084-y) or ferments (e.g., https://doi.org/10.1016/j.mimet.2016.10.020) are employed. Both specific and non-specific LAMP products can give high Tm values (e.g., https://doi.org/10.1093/nar/gkaa099). As a response to the Reviewer question, we have provided melting curves for specific (C. sepedonicus) and non-specific (generated at large DNA loads and the extended amplification time for Dickeya and Pectobacterium species) LAMP products as the new Figure S4 in the revised manuscript.
Could the authors quantify the observed 2–3× delay in amplification in the absence of loop primers—was this based on threshold time (Tt) or endpoint fluorescence intensity?
We conducted a few amplifications without loop primers, just to produce some amount of LAMPlicons in order to test whether LAMP products generated with and without loop primers can be recognized by the Cas12a/gRNA complex. In the absence of loop primers, we observed a delay in a sigmoid-like rise of fluorescence. So, it is about amplification kinetics and based on “threshold time”, in particular, on values of the quantification cycle, provided by the PCR machine software. We specified it upon revision (line 172) in response to the Reviewer’s question.
Were there any differences in amplification efficiency or product yield when loop primers were excluded?
In PCR, amplification efficiency means how effectively amplicons are copied in each cycle. So, in the rigorous sense, we cannot say anything about amplification efficiency in our case, unless we use this term as a substitute for amplification kinetics. The kinetics of amplification was much slower in the absence of loop primers, while the observed maximal values of end-point fluorescence were close. It may imply that yield of LAMP products is similar in both cases. In contrast to amplification kinetics which slowed dramatically, the differences in product yield appear rather minor, if exist at all. Yet, due to sparse data, we cannot make a firm conclusion on product yield in the presence and absence of loop primers for our particular case.
Could the authors provide experimental data or references confirming that pyrophosphate accumulation during LAMP is sufficient to sequester Mg²⁺ and inhibit Cas12a activity under their conditions?
No, we cannot. In fact, the Reviewer’s question led us to re-think our “working hypothesis” on the role of pyrophosphate in the observed inhibition of Cas12a trans-activity. The Reviewer is absolutely right that pyrophosphate cannot sequester magnesium ions in the Cas12a reaction mixture. The amount of pyrophosphate could hardly be equal to half the amount of magnesium ions in LAMP reaction mixture, the majority of pyrophosphate is already in complex with magnesium, and there will be no enough free pyrophosphate to bind magnesium ions in the Cas12a reaction mixture.
When we run into the problem with Cas12a showing no trans-cleavage activity in 50 µL reaction mixture combined with 10 µL completed LAMP reaction, conducted under a layer of mineral oil, we first varied the added volume of LAMP reaction to find a solution to this problem. Indeed, trans-cleavage activity gradually decreased to zero when the added volume increased from 1 to 10 µL. The first thought was about components of LAMP reaction mixture. So, we tested samples containing 1 µL of completed LAMP reaction plus 10 µL of the full LAMP reaction mixture (includes target, primers, dNTPs, buffer, etc., but not subjected to incubation at 65°C). The cleavage kinetics were practically identical to those obtained with 1 µL of the completed LAMP reaction alone. The second thought was about mineral oil. Yet, the kinetics were identical, whether mineral oil was added or not. Thus, we concluded that some bypass products of LAMP reaction could be a reason. Pyrophosphate was the first we have thought of. Since pyrophosphate is able to bind magnesium ions, we elevated the magnesium chloride concentration and, bingo, it has turned to work out. Ironically, the erroneous premise led us to the practical solution of the problem we run into.
We have rewritten the relevant paragraphs in the Results section upon revision to address the Reviewer question (lines 297-324 of the revised manuscript). We are unable for the time being to suggest a reasonable hypothesis which would account for the observed inhibition of Cas12a trans-cleavage activity and its restoration by elevating magnesium chloride concentration. Nonetheless, we have found a simple technical solution (though by a chance) which allowed us to continue experiments and demonstrate that the loop-targeted PAM-independent Cas12a analysis of LAMP products is compatible with a format of a single test tube assay and non-instrumental (naked-eye) detection.
Was the Cas12a activity restored immediately upon Mg²⁺ addition, or was an incubation period required for complex formation?
In the present setup, magnesium chloride was added during the preparation of Cas12a reaction mixtures, prior to the Cas12a/gRNA complex formation, both for commonly used (6 mM) or elevated (9-18 mM) Mg²⁺ concentrations. Since, in that case, we were interested first of all in evaluating compatibility of the loop-targeted LAMP/Cas12a system with a format of a single test tube assay and naked-eye detection, such procedure seemed to be more practical than adding magnesium ions to their amount already present in the reaction mixture. However, the Reviewer is absolutely right that, by looking at how the Cas12a activity is restored if magnesium ions are added in different ways, one could narrow the options for mechanistic explanation of the observed phenomenon, at least. In any way, further efforts will be required to get mechanistic insights that, we believed, is out of scope of the present study where the major focus was on a new strategy to circumvent the PAM requirement by targeting single-stranded loops of LAMPlicon.
All changes made in the manuscript are highlighted by yellow.

Reviewer 4 Report
Comments and Suggestions for Authors
The manuscript titled “PAM-independent Cas12a detection of specific LAMP products by targeting amplicon loops” describes a method based on the selective recognition of specific products of loop-mediated isothermal amplification (LAMP) using Cas12a nuclease, without the need for a protospacer adjacent motif (PAM) in the sequence of LAMP amplicons. The strategy relies on the presence of single-stranded DNA loops in LAMP amplicons and the ability of Cas12a to be activated through binding of guide RNA (gRNA) to single-stranded DNA, in the absence of a PAM sequence.
The aim of the present work was to test whether a truly PAM-independent LAMP/Cas12a detection system can be developed by targeting single-stranded loops of LAMPlicons with gRNA.
The feasibility of this approach was demonstrated using Clavibacter species – an important group of bacterial plant pathogens that cause harmful diseases in agricultural plants including potato, tomato, maize, wheat, legumes, and some others.
For Clavibacter, the sensitivity of the developed PAM-independent LAMP/Cas12a system was compared to that of LAMP alone, while the overall selectivity of detection was improved by the Cas12a analysis of LAMP products.. It has been demonstrated that the LAMP/Cas12a detection system can be optimized by carefully designing gRNA to selectively distinguish Clavibacter sepedonicus (potato ring rot - causing agent) from other Clavibacter species based on single-nucleotide substitutions in the targeted LAMPlicon loop.
The proposed loop-based Cas12a analysis of LAMPlicons is compatible with a single-tube assay format with the option for naked-eye detection. These findings expand the range of approaches for designing PAM-independent LAMP/Cas12a detection systems, with the potential for on-site testing and can be incorporated into State-regulated standards for seed potato health assay.
The manuscript is well-written and describes a carefully done research work of high importance for plant pathology field.
Author Response
The manuscript titled “PAM-independent Cas12a detection of specific LAMP products by targeting amplicon loops” describes a method based on the selective recognition of specific products of loop-mediated isothermal amplification (LAMP) using Cas12a nuclease, without the need for a protospacer adjacent motif (PAM) in the sequence of LAMP amplicons. The strategy relies on the presence of single-stranded DNA loops in LAMP amplicons and the ability of Cas12a to be activated through binding of guide RNA (gRNA) to single-stranded DNA, in the absence of a PAM sequence.
The aim of the present work was to test whether a truly PAM-independent LAMP/Cas12a detection system can be developed by targeting single-stranded loops of LAMPlicons with gRNA.
The feasibility of this approach was demonstrated using Clavibacter species – an important group of bacterial plant pathogens that cause harmful diseases in agricultural plants including potato, tomato, maize, wheat, legumes, and some others.
For Clavibacter, the sensitivity of the developed PAM-independent LAMP/Cas12a system was compared to that of LAMP alone, while the overall selectivity of detection was improved by the Cas12a analysis of LAMP products.. It has been demonstrated that the LAMP/Cas12a detection system can be optimized by carefully designing gRNA to selectively distinguish Clavibacter sepedonicus (potato ring rot - causing agent) from other Clavibacter species based on single-nucleotide substitutions in the targeted LAMPlicon loop.
The proposed loop-based Cas12a analysis of LAMPlicons is compatible with a single-tube assay format with the option for naked-eye detection. These findings expand the range of approaches for designing PAM-independent LAMP/Cas12a detection systems, with the potential for on-site testing and can be incorporated into State-regulated standards for seed potato health assay.
The manuscript is well-written and describes a carefully done research work of high importance for plant pathology field.
We thank the Reviewer for positive appraisal of our work.

Round 2
Reviewer 1 Report
Comments and Suggestions for Authors
Many thanks to authors for their thoughtful comments and careful corrections of the manuscript. Several questions from the first review and the readability of the manuscript have been increased. However, several issues still require more attention and probably editing.
- While several complex sentences were edited and simplified, the overall writing style of the manuscript remains difficult to read. Specifically, consequent long sentences make the text cramped and could be divided or changed.
- Supplementary file seems to be identical to the first submission and does not contain changed data mentioned in the author’s reply.
- Concerning end-point measurement, the authors point of view is fully valid taking into account the proof-of-concept nature of the study. However, kinetics itself does not allow to fully understand the true difference in fluorescence within the used reaction conditions. For instance, prolonged cleavage time leads to color change in NTCs and reactions with non-targeted species. Sensitivity of naked-eye detection is obviously different from specialized equipment, and even two-fold or similar degree of difference can be hard to detect visually, especially, for certain color patterns. This effect is well-known for LAMP and was reported for fluorescent, pH-dependent and other type of dyes. In that light, fluorescent curves can demonstrate the feasibility of various conditions for visual detection of LAMP results and these curves need to be presented together with kinetic assessment at least in the Supplementary.
Author Response
Many thanks to authors for their thoughtful comments and careful corrections of the manuscript. Several questions from the first review and the readability of the manuscript have been increased. However, several issues still require more attention and probably editing.
We are very grateful to the Reviewer for careful reading of the manuscript, interesting and valuable discussion, and useful remarks and recommendations.
While several complex sentences were edited and simplified, the overall writing style of the manuscript remains difficult to read. Specifically, consequent long sentences make the text cramped and could be divided or changed.
We are thankful to the Reviewer for pointing out the specific problem with the writing style. We went through the manuscript once again and divided or rephrased the long sentences where possible. The corrections made are highlighted in yellow color.
Supplementary file seems to be identical to the first submission and does not contain changed data mentioned in the author’s reply.
As far as we understand, the Reviewer means Table S3 where we had to present characteristic amplification times recalculated based on values of quantification cycles (Cq). We have double-checked the supplementary file presented with the revised manuscript. Table 3 contains recalculated data. For Reviewer convenience, we are providing Tables 3 from the initial and revised versions of Supplementary Materials below.
The initial version:
The revised version:
Concerning end-point measurement, the authors point of view is fully valid taking into account the proof-of-concept nature of the study. However, kinetics itself does not allow to fully understand the true difference in fluorescence within the used reaction conditions. For instance, prolonged cleavage time leads to color change in NTCs and reactions with non-targeted species. Sensitivity of naked-eye detection is obviously different from specialized equipment, and even two-fold or similar degree of difference can be hard to detect visually, especially, for certain color patterns. This effect is well-known for LAMP and was reported for fluorescent, pH-dependent and other type of dyes. In that light, fluorescent curves can demonstrate the feasibility of various conditions for visual detection of LAMP results and these curves need to be presented together with kinetic assessment at least in the Supplementary.
We completely agree with the Reviewer that differences in intensity of fluorescence are not easy to estimate by eye in general, even if they are two-fold. For the particular case of ROX-labeled reporters, the change in fluorescence intensity results in a color transition from blue to purple, visible under daylight. From our experience of working with ROX-labeled reporters, we got impression that the change of color from blue to purple occurred when fluorescence intensity crossed the values of about 10,000 to 12,000 arbitrary units. The color transition was rather sharp than gradual.
The representative kinetic curves for ROX-MR-8 cleavage by the activated Cas12a nuclease are provided as Figure S8 in the newly revised version of Supplementary Materials, as a response to the Reviewer request. The relevant discussion is given in lines 374 to 383 of the main text of the manuscript and highlighted in blue color.

Round 3
Reviewer 1 Report
Comments and Suggestions for Authors
Many thanks to authors for their detailed replied and careful corrections of the manuscript. No further corrections are necessary and the manuscript can be accepted for possible publication in the present state.